# Wear Mechanisms of the Forging Tool Used in Pre-Forming in a Double Forging System of Truck Parts

**DOI:** 10.3390/ma16010351

**Published:** 2022-12-30

**Authors:** Janusz Krawczyk, Aneta Łukaszek-Sołek, Łukasz Lisiecki, Tomasz Śleboda, Marek Hawryluk

**Affiliations:** 1Faculty of Metals Engineering and Industrial Computer Science, AGH University of Science and Technology, Av. Mickiewicza 30, 30-059 Krakow, Poland; 2Department of Metal Forming, Welding and Metrology, Wroclaw University of Science and Technology, Lukasiewicza Street 5, 50-370 Wroclaw, Poland

**Keywords:** hot forging, tool wear, surface layer, white etching layer, tribology

## Abstract

Tool life in plastic forming processes is a problem of the utmost importance as it significantly affects the cost of production. Hot forging with hammers and mechanical presses is an example of the technological process in which the load on tools is extremely high and, consequently, the lifetime of tools is short. Considering, additionally, that this applies to large-scale production, from an economic point of view, the key issue will be to extend the tool life, make an accurate prediction of the number of parts that can be forged before the replacement of dies is necessary, and develop a system for quick tool changeover. Initially, however, it is necessary to understand the causes of excessive tool wear, which may lie in phenomena occurring at the level of microstructure. The aim of this article was to outline an example of the coexistence of multiple wear mechanisms in hot forging dies. For the modified chemical composition, the microstructure examinations were performed in selected areas of the tool. The research has revealed the causes of cracks in tools and some irregularities in the preparation of tools for production process.

## 1. Introduction

The problem of tool wear is extremely important from an industrial point of view. It is assumed that the cost of tools is from 8 to 15% of the cost of the manufacturing process [1,2], and in some cases, when the tools are heavily loaded, e.g., when the forged parts are of complex shapes, this cost may increase even up to 50% [2,3]. Therefore, this issue is frequently raised by research communities around the world. Both tool wear forecasting and the extension of a tool’s lifetime have an impact on the correct course and optimization of production. That is why knowledge and understanding of wear mechanisms are so important and mark the starting point for further efforts to improve the life of tools [4,5,6,7].

The durability of hot forging tools is also most often determined by the number of cycles for which the tool met the set requirements, e.g., dimensional, mechanical, and quality, and this is determined primarily by the contact of the tool surface with the processed stock, by the mechanical and thermal loads recurring in cycles, and by the intense abrasive wear proportional to the number of forgings produced [8]. Depending on the size, shape intricacy, and quality requirements imposed on forgings and the type of stock material, the durability of hot forging tools ranges from 3000 to 25,000 cycles. The high temperature forming process entails demanding working conditions to the forming steel dies. The main damage mechanisms leading to die failure in hot forging processes are abrasive and adhesive wear, mechanical and thermomechanical fatigue, plastic deformation, and oxidation, and in addition, during operation, the working surface of tools designed for hot work is exposed to: high temperature effects, cyclic temperature changes, cyclic stress changes, dynamic mechanical loads, and intense wear [2,9,10,11]. According to the statistics, 70% of forging dies are taken out of service due to wear, 25% due to mechanical fatigue, and only 3–5% of the forging dies failed due to thermal cracking and plastic deformation [1,12]. In the hot die forging process, the temperature of die inserts is an equally important factor, partly responsible for the creation of proper tribological conditions [13]. The material at the working surface is exposed to the highest temperatures during die operation. A special case of microstructural changes resulting from the tribological contact between the surface layer of parts made from iron-based alloys is the phenomenon of the formation of the so-called white layer [14]. The white layer is a difficult-to-etch, thin surface layer formed on the material (less than 20 μm thick), characterized by high hardness (over 1000 HV) and high brittleness. It is the product of very rapid heating to high temperatures (the temperature range of austenitic transformation), accompanied by severe deformation (lattice defects) and followed by immediate cooling, usually to room temperature. This layer can radically change the wear mechanism, operating not only in tools but also in machine parts. Most research works in this field are focused on surfaces after machining [15,16,17,18,19], especially steel surfaces hardened by heat treatment (quenching and low tempering) and subjected to machining. An issue widely discussed in the literature is the formation of a white layer on the running surfaces of railway rails [20,21,22,23,24,25], but there is only limited information on the occurrence of this layer on the surface of tools used in the processes of plastic forming of metals. However, the harsh operating conditions of tools such as metallurgical rolls [26] or forging dies [27] will undoubtedly favour the formation of a white layer, especially when the proposed theories of the mechanism of its formation are taken into account [28].

Zener et al. concluded in their work [29] that the white layer component is primarily unreleased martensite formed from austenite formed by heating above Ac3 temperature. Such a theory is supported by the increase in the proportion of residual austenite in the white layer with respect to the base material [30]. Given the observations indicating very fine grains in the white layer area, it is assumed that its formation is related to very intense plastic deformation, dynamic recovery, and the recrystallization processes [31,32]. The first factor in the formation of the white layer is the heating of the surface layer of the material to high temperatures, which can occur, for example, due to high frictional forces between the forging die and the workpiece material. The formation of the white layer is often accompanied by heating of the layer immediately below it to temperatures close to the Ac1. If the raw material was hardened and tempered, then the layer lying under the white layer is tempered again very high and etched intensively; this is an area with a different structure, the so-called dark layer. The hardness of the dark layer is less than the hardness of the white layer and often even less than the hardness of the raw material. The second phenomenon that causes the formation of the white layer is a strong plastic deformation that causes distortion/deformation of the material structure (at a high temperature corresponding to the existence of austenite). The third condition for the formation of the white layer is the rapid cooling of its area immediately after the completion of its deformation (e.g., between bars of forging, when the shaped material has been removed from the die and the tool is exposed to ambient temperature or with the contact of the new material with the tool). As a result of such cooling, very strongly deformed material can remain in the white layer area as nanocrystalline. For this reason, the thickness of the white layer is always very small. The white layer increases frictional forces; nevertheless, its effect depends on the occurrence of cracks in the white layer as well as on frictional conditions [33,34].

The aim of the present study was to discuss the coexistence of many wear mechanisms characteristic of the hot forging process. To achieve this goal, a demonstration was made using a selected die insert, for which a comprehensive set of tribological wear mechanisms was practically identified.

## 2. Materials and Methods

The test material was a die insert for pre-forming made from the modified (based on the chemical composition analysis) X37CrMoV5-1 (1.2343) tool steel for hot work (Table 1), having the impact strength in the range from 12 J to 33 J. The die (Figure 1) was designed for operation in the double multiple hot forging process of truck parts using an LZK 1000 mechanical press of 1000 T capacity and ram stroke of 220 mm/min. The die insert was mounted in a specially designed holder. The range of operating temperatures of the pre-forming die insert was 200–250 °C, which required its preheating before the production process to avoid the risk of brittle fracture when the process was started. The die inserts were lubricated with a 10% aqueous graphite solution. The forging process of the workpiece included swaging, pre-forging, and finished forging. The charge was heated to 1100 °C. The billet material was 42CrMoS4 structural steel.

Figure 1a shows a geometric model of the designed die insert for pre-forming in a double forging system. Considering the complexity of the multiple forging process, the possible risk of deformation of the die inserts and the shape of forgings (intricate and compact with variable cross-sections), it was very important to properly design the die cavity. Thus, compared to the post-forming die insert, the pre-forming insert was characterized by largely increased fillet radius and depth but reduced width of the cavity and properly matched inner flash gap. Moreover, to obtain easier degradation of material, the insert was made with a low degree of plastic deformation, which resulted in its higher brittleness and lower anisotropy of the tool material properties. Based on previous experience and analysis of modelling results, the tool life was estimated at 5000–6000 blanks. Figure 1b shows the insert in post-operation period, i.e., after withdrawal from the production process due to wear and cracks. In industrial practice, the wear of die inserts demands retooling of the press, which means downtime in the production process. During pre-forming, the machine is also expected to withstand the most critical stresses. 

In order to perform an analysis, the following research methods were applied:macroscopic analyses with a measurement of the degree of wear/loss of the material on the tool’s working surface by using a 3 d scanning method with the measuring arm ROMER Absolute ARM 7520si integrated with a RS3 scanner, as well as a comparison of the scan geometry with the CAD models;chemical composition analysis was determined by An X-ray fluorescence analyser (Twin-X, Oxford Instruments, Witney, Oxon);the microstructure of the subsurface layer of the used tool was examined using an MMT 120BT optical microscope with an HDCE-X3 camera and a ThermoFisher Phenom XL scanning electron microscope (the die insert was cut along its shorter side to make test specimens; grinding and polishing on a grinder-polisher Struers 350, the tool’s samples were etched with 4% nital—94 mL H_2_O, 4 mL HNO_3_, 2 mL HF);the hardness measurements were made with a Struers Duramin 40 M1 hardness tester;the numerical modelling was carried out in the three-dimensional state of deformation in the hot forging process in QForm software. The purpose of the simulations was to determine the temperature distribution, effective stress, and strain for the die insert in the hot forging operation in terms of tool life estimation. Boundary conditions for the simulation of the forging were selected based on the analysis of the industrial process.

## 3. Research Results and Discussion

### 3.1. Chemical Composition

Chemical analysis of the examined pre-forming die insert indicates the use of the modified X37CrMoV51 (1.2343) steel (Table 1). Detailed analysis of the chemical composition shows that the carbon content is about 0.11% higher than the maximum level. In the case of tool steel for hot work, this may have an impact on the decrease in plastic properties. Carbon content increasing in the iron alloy increases some mechanical properties, e.g., hardness, but then the brittleness of the steel also increases, which, in the case of forging tools, results in highly undesirable consequences. High carbon content affects the hardenability of steel, which should be taken into account when planning heat treatment. Additionally, when carbon content increases in the alloy, the result may be the formation of a large amount of carbides or residual austenite after quenching. The silicon content is approximately 4.5 times lower than the average content provided for in the EN ISO 4957: 2018 standard. The content of this element at such a low level may adversely affect the heat resistance of the tool as well as its hardness after tempering. The content of manganese is in the upper standard range. Manganese in tool steels increases hardenability, but also has a negative effect, increasing the steel tendency to overheating. The molybdenum content is around 2.29%, which, by almost 0.8%, exceeds the upper limit specified in the standard. A high content of this element increases the alloy hardenability and abrasion resistance. Additionally, it delays the thermal softening of steel at elevated temperature, which helps to maintain the correct tool geometry. Sulphur and phosphorus are in the standard range, so they have no adverse effect on the strength properties of the material. The content of chromium is below the minimum level, which may have a negative effect on the material properties, as this element improves both hardenability and depth of hardening. The vanadium content is just above the maximum level. Proper vanadium content guarantees the fine-grained structure of steel. It increases the steel resistance to overheating, improves the abrasion resistance, and delays the decrease in hardness caused by the tempering effect of a temperature up to 600 °C.

### 3.2. The Macro Analysis by 3D Scanning Method and Morphology of Surface Layer

To determine and verify the life of forging tools, taking as an example the pre-forming die insert operating in a dual system, the macrostructure was analysed. The results of 3S scanning are presented in Figure 2.

The greatest wear indicating a large loss of material is in the middle part between the blanks and is locally even above 3 mm. This wear is the result of intensive flow of the charge material during upsetting. The scan image can also show a crack that appeared in the place of the smallest cross-section of the tool, most likely as a result of cyclical high pressures and temperature gradients.

Other areas where a large loss of material can also be observed are the places where the forging material flows through the bridge into the flash groove. In these areas, typical abrasive wear dominates; the loss of material in the normal direction is in the range of 1.3 to 1.8 mm.

Furthermore, the microstructure of the subsurface layer was examined in the areas marked in Figure 3. 

The analysis of the first area (Figure 3—area 1a), left idle during forging (except for the outer flash gap), showed that, in this area, the tool was not subjected to any thermo-chemical treatment (Figure 4a). A thermo-chemical treatment, such as, e.g., nitriding, would leave a visible layer on the tool with a structure different from the parent material. Microstructure examinations indicated the presence of tempered martensite and alloyed carbides. A thin layer of material undergoing plastic deformation is also visible. This deformation was most likely caused by the machining process performed on the tool.

A fragment of the central part of the die insert was also examined (Figure 3—area 2a,3a). This is an area located in the central part of an inner web separating the two forged blanks and, more precisely, a part of the inner wall of one of the die cavities and a web separating these cavities (Figure 4b). In the initial stage of forging, loading of this area is the most critical. The examined fragment was divided into two main areas, marked I and II. In area I, the consequences of the abrasive wear are clearly visible. There are numerous grooves formed by metal flowing towards the outer wall of the die cavity. Here, the web separating the die cavities also suffered from heavy abrasive wear. This may indicate a very intense flow and movement of a large amount of the stock material striving to properly fill the die cavity. On the other hand, area II is the place where the stresses disengage the tool with the resulting minor abrasive wear in a more remote part of the die cavity. During hot die forging, the temperature of the tools changes in cycles, which contributes to thermo-mechanical fatigue and fatigue cracking and, as a consequence, to damage of the tool. The temperature of the tools increases as a result of contact with hot metal, conversion of the work of plastic deformation into heat, and friction during subsequent operations of the forming process (the average post-forming temperature on the surface of an insert may exceed 600 °C). The tool temperature drops to about 100 °C or less as a result of the use of an appropriate type and amount of lubricant. In the area of an inner web (Figure 2, Figure 3 and Figure 4a), the observed predominant wear mechanism was also high-temperature oxidation (Figure 4c). The formation and abrasion of successive oxide films in the top layer left numerous irregularities on the surface of the die insert. On the other hand, the long exposure of the tool material resulted in its tempering.

The morphology of the top layer in the area where the die profile changes into an outer flash gap (Figure 3—area 3b1,3b2) indicates volumetric fatigue wear (Figure 5). A characteristic feature of this type of wear is plastic deformation of the die cavity as a result of repeated friction at high pressure and temperature and also cracks formed by the forged material pushing with great force against the walls of this cavity. The accumulation of resulting contact stresses caused the nucleation of microcracks, which turned into macrocracks during further operation of the tool which were penetrating deeper into the material. The “closure” of the nucleation site, characteristic for this type of cracks, was due to further plastic deformation of the top layer during tool operation. The genesis of such cracks may be mechanical fatigue, additionally enhanced by thermal fatigue.

### 3.3. The White Layer

The higher the pressure/stress on the contact surface, the closer the contact of the tool surface, with the steel heated up to the point of the austenite formation and the most intense metal flow heating the tool surface layer occurring on the die cavity wall in the area of the transition of this wall into an outer flash gap and mainly in the area of the web present in the gap. This is due to the fact that, in this place, the forged material undergoes the most severe plastic deformation and, additionally, it moves along the tool’s surface at the highest speed. Therefore, as a result of tribological interaction and the exceeded temperature of austenitic transformation, new phases may arise in the top layer of the tool. A phenomenon inherent to this transformation will be the formation of the so-called white layer [35,36,37]. The white layer is a thin, super-hard, very brittle and difficult-to-etch layer that radically changes the wear mechanism of tools. Metallographic examinations performed on the surface layers of the examined samples (Figure 3—area 1a, 3b1, 4a) revealed the presence of the difficult-to-etch microstructures. Considering the operating conditions of the tool and the wear mechanisms predominant in these areas, the formation of a white layer seems to be quite natural (Figure 6).

The difficult-to-etch areas have developed in this region (indicated by red arrows in Figure 6a). Their “tunnel-like” shape indicates the abrasive character or wear. At high pressure and temperature, when the processed material was flowing out of the die, it was moving in these “tunnels”, deforming, at the same time, the surface of the tool. Comparing the geometry of the difficult-to-etch areas and their morphology, shown in Figure 6a–c, it can be stated that plastic deformation of the top layer in the areas of the white layer was directional. Using the cross-section perpendicular to the web edge in the examinations shown in Figure 6a and the cross-section parallel to this edge in the examinations shown in Figure 6b,c, it was possible to indicate the direction of the plastic flow of material in the area of the white layer. The tool material flow was plastic and concurrent with the direction of the forged material’s plastic flow. The case of the white layer formation presented in Figure 6d shows that a new white layer was formed in the place of the previously existing white layer. This is indicated by the presence of a boundary between the white layer and the area, also difficult-to-etch, lying underneath this layer. Only by penetrating deeper inside can one see the parent material, also etched in a light colour. The thermal effects accompanying the formation of the new white layer in the part of the area occupied by the previously existing white layer resulted in the tempering of the “old” white layer material. The most intense tempering of the previously formed white layer has occurred in the area closest to the newly formed white layer. The result was the occurrence of a dark line separating the two light-colour etched areas. 

The white layer, apart from being a difficult-to-etch phase, is also characterized by a very high hardness (>1000 HV). Its formation is due to a rapid increase in the temperature of the surface layer to the range of the austenite formation, followed by equally rapid cooling and the resulting martensitic transformation. This process is accompanied by severe deformation of the crystal structure. To confirm the existence of a white layer on the tool’s surface, hardness measurements were made in area 4a (Figure 3), using, for this purpose, a Struers Duramin 40 M1 hardness tester. Hardness was measured by the Knoop method, which enabled measuring changes in this parameter at a small distance from the surface of the die insert. Several measurements were taken. From the analysis of the measurement results (Figure 7), it can be concluded that as a result of tribological wear of the die insert at high operating temperatures of the tool, a white layer was formed on the surface of this tool. The occurrence of the white layer in the top layer was additionally confirmed by the fact that in the examined area of the top layer, the temperature of the tool was higher than the temperature of the austenite formation (840 ÷ 875 °C). 

Detailed analysis of the results of hardness measurements allowed for distinguishing five layers of different hardness in the examined material. The formation of these layers is the result of differences in the deformation and/or heating rate of the tool during operation. These are the following layers:White layer, formed as a result of rapid cooling of the heavily damaged austenite;Material with less severe plastic deformation than the white layer in the area heated during operation to the range of the austenite formation;Austenitized layer free from plastic deformation and with a large amount of residual austenite after cooling (apparently, hardness of this material is lower than that of the parent material);Dark layer, i.e., highly tempered parent material of the die;Parent material of the die.

### 3.4. Nucleation of Fatigue Cracks as a Consequence of the Formation of White Layer

The white layer, characterized by very high hardness and brittleness, is composed of martensite formed as a result of cooling the heavily damaged austenite. As a microstructure with the highest specific volume, martensite increases the level of stresses between the surface layer and the base material, thus facilitating the development of cracks. Therefore, the formation of a white layer on the surface of the tool explains the formation of cracks, shown earlier in Figure 6b,c. Particularly, Figure 6b shows very clearly the path of the cracks, which at the beginning are running parallel to the layer but later start propagating deeper into the material (the crack path is indicated in the drawings with dashed red arrows). Figure 6c shows the exposed layer of material caused by further development of the crack along the white layer that has broken off during prolonged operation of the tool, allowing for the formation of a new layer and continuation of the die surface’s degradation process. A similar mechanism of the defect formation operates in the part of the white layer marked with a yellow dotted line in Figure 6b.

### 3.5. The Effect of White Layer Formation on Adhesive Wear

Sticking of material is the phenomenon commonly coexisting with the formation of a white layer on the surfaces of hot forming tools. An interaction between these phenomena is possible mainly due to the formation of an easy adhesive bond between the austenitized layer resting on the tool and austenite present in the wrought material [38]. The examination of the subsurface layer in a fragment of the flash web in a sample, and more precisely, on the fillet radius at the die cavity’s transition into a web, indicates sticking of the forged material to the die surface (Figure 8). This is the result of the occurrence of adhesive wear with its characteristic feature, which is the formation of a permanent bond between the tool’s surface and the forged material during plastic flow of this material over the die wall. The adhesive bond is stronger than the cohesive bond formed with the forged material. With further flow of the material, the decohesion, or the detachment of some fragments of the forged stock, occurs, and the detached fragments are next transferred to the surface of the die insert. Figure 8 shows how clear the difference is between the sticking material and the top layer of the die insert.

### 3.6. Characteristics of Crack Development in the Central Part of the Die Insert

In the central part of the tool, the die insert has a crosswise running crack, which is a continuation of the crack that occurs in the remaining parts of the tool. The fragment of the tool shown in Figure 9a (Figure 3—area 5a) covers the area of the two die cavities, an inner web separating these cavities, and an outer web. It is worth noting that in this place, the crack (Figure 1), whose direction was so far parallel to the side walls of the die cavity, has changed its plane to perpendicular. This may be due to the fact that the examined fragment is located in the central part of the die cavity, exposed to the effect of the most critical loads in the initial stage of the forging process. This area is particularly vulnerable to wear, as the flowing material is confined on several sides by the walls of the die, with the result that, locally, there is an increase in stresses, and the resultant direction of the stresses leads to loosening of the tool fixing. Figure 9a shows macro pictures of the examined areas. Due to the crack from area A, the area shown in Figure 9b was analysed by scanning electron microscopy. This fragment represents the corner of the impression die that broke during operation. A spall of material from the working surface can be observed on a section of the die (I). This spall then transitions into a localised coarsening of the tool to again transition into a slightly smaller spall. This part is an area of delamination resulting from thermal fatigue. Another area, marked II, is the result of fatigue at the point of change in the shape of the die, which has fractured along one plane almost across the entire tool. Figure 9c,d shows the transition between the working surface and the fatigue–thermal delamination. The fatigue striations are arranged perpendicular to the chipping edge. In contrast, the transition between thermal fatigue delamination and fatigue crack II can be observed in Figure 9e.

The crack has the character of delamination beneath the surface, involving Hertzian stresses, with visible striations indicating crack nucleation sites. The area B shows a fragment of the breaking crack (Figure 9f) that extends across the die insert. This is a secondary effect of the occurrence of earlier cracks, not related to the crack running in a “from-the-surface” direction. Additionally, in this place, there is a fatigue crack running perpendicular to the breaking crack. Traces of abrasive wear are also visible, with characteristic grooves running parallel to the edge of the breaking crack. This is related to the initial stage of deformation of the swollen material that flows towards the outer walls of the die cavity. The fatigue crack with the nucleation site located deep in the sample “meets” the edge of the breaking crack that propagates across the tool. The “meeting place” is the resulting notch (Figure 9g,h) and change in the plane of the breaking crack. Additionally, slight chipping of the material occurs in the place where the notch has been formed (Figure 9i,j). To adequately characterize the crack development in the die material, the microstructure was examined in the areas of the secondary cracks (Figure 9a—area C). Figure 10 shows the secondary crack developed on the sample in the central part of the die insert; visible are the site of the crack and the secondary crack (indicated by arrows, respectively). 

The crack path is both transcrystalline and intercrystalline, also making use of the needle-like structures of tempered martensite and the interfacial structures of carbides and matrix.

### 3.7. Numerical Analysis of Die Insert

The numerical modelling of hot forging was performed using QForm 3D commercial software based on FEM. The temperature for forging the billet was 1100 °C. An initial temperature of the tools was assumed as 250 °C. The time of the billet transportation to the die cavity was 2 s, and the time of billet cooling in a die cavity before forging was 1 s. A graphite–water emulsion, having a friction factor of 0.4, was assumed as a lubricant in numerical modeling.

The forging process analyzed involves the intensive movement of charge material under high unit pressure. Friction on the tool surface is a natural cause of wear. The layer of lubricant at high unit pressures can break, exposing the surface of the charge covered with hard particles that increase tool wear. The obtained results of the numerical modelling of the die insert are presented in Figure 11. Figure 11a shows the temperature distribution on the tool surface. The maximum temperature reached during pre-forging is about 650 °C at the outer bridge. The bridge separating two die cavities is in contact with the heated material from the very beginning of the process, as a result of which it reaches high temperatures. The highest values are reached near the smaller corners of the die cavity, on the outer side. The lowest values are observed on the outer side of the die cavity bridge. This is due to the material flow inside the die cavity strongly affecting the tool. Taking into account the distributions of effective strain and effective stress, it is possible to observe the highest values in the corner at the area of die cavity–outer flash groove. Comparison of the results of numerical modelling with the results of 3D scanning confirms the high wear of the insert in the area of the inner flash and at the transition of the die cavity into the flash groove.

## 4. Conclusions

This article presents the results of wear tests for a forging tool used in the hammer process for pre-forming in a double forging system. The analyses carried out showed that the implemented technology is complicated in the industrial implementation of the process and requires changes in the key parameters of the technological process in order to eliminate premature tool wear. Optimization of the geometry of the working patterns of the tools and the use of alternative material solutions and surface treatments using surface engineering methods should also be considered. On the basis of the conducted research, the most important conclusions can be formulated: -The tool material in the area of the surface layer remaining idle during operation shows only minor changes compared to the parent material and, as such, can be considered a proper (reference) material for the analysis of changes that occur in the parent material as a result of its active participation in the production process.-The geometry of the tool surface also has a significant impact on the formation of fatigue cracks—the dislocation effect. Based on the 3D scanning, a large loss of material occurs in the middle part between the blanks and is locally even above 3 mm. This wear is the result of intensive flow of the charge material in these areas.-Tribological interaction (including the state of strain and thermal interaction) results in the formation of the so-called white layer. The morphology of the white layer is directly related to the intensity and direction of the forged material flow.-The formation of a white layer, and, more precisely, the austenitization of the tool surface layer associated with the formation of a white layer, favours the formation of sticking films (adhesive wear). Fatigue cracks can be initiated by the formation of a white layer.-The morphology of the white layer indicates the formation of a hard layer on a softer intermediate layer lying between the hard layer and the parent material; this will facilitate chipping of the layer and, consequently, local increase in the wear rate.-On a macro scale, the development of cracks depends on the state of stress, and on a micro scale, it depends on the structure of tempered martensite and carbide morphology.-Directions for further research, allowing us to obtain additional information about mechanisms of fractures in the working area of forging die, should be investigated based on numerical simulation of the process.-The dominant wear mechanism depends on the location of the die cavity’s surface area (Figure 12).

## Figures and Tables

**Figure 1 materials-16-00351-f001:**
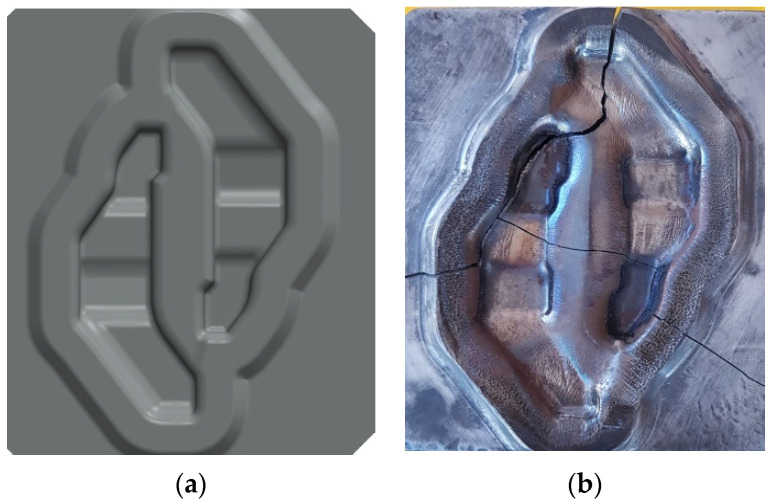
The view of: (**a**) geometric model of the designed die inserts for pre-forming in a double forging system and (**b**) the insert in post-operation period.

**Figure 2 materials-16-00351-f002:**
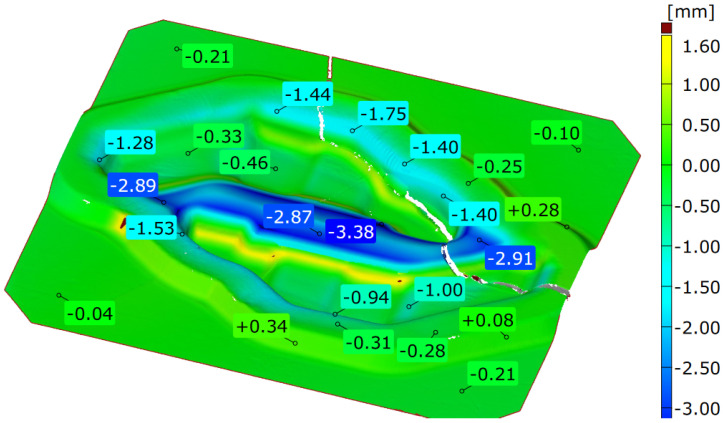
The results of the scan analysis of the used tool.

**Figure 3 materials-16-00351-f003:**
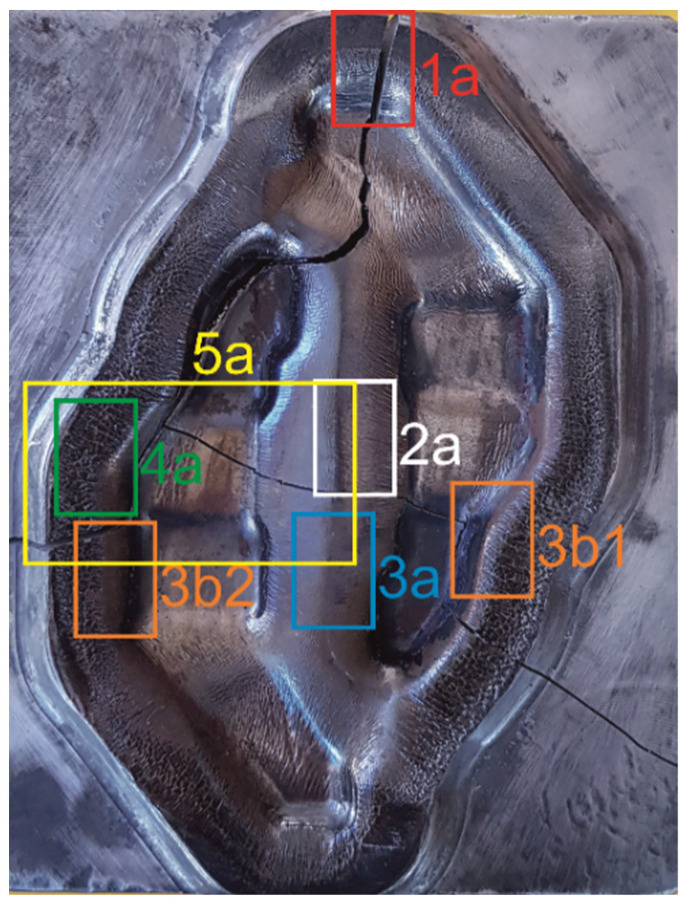
The pre-forming die insert with areas selected for analysis.

**Figure 4 materials-16-00351-f004:**
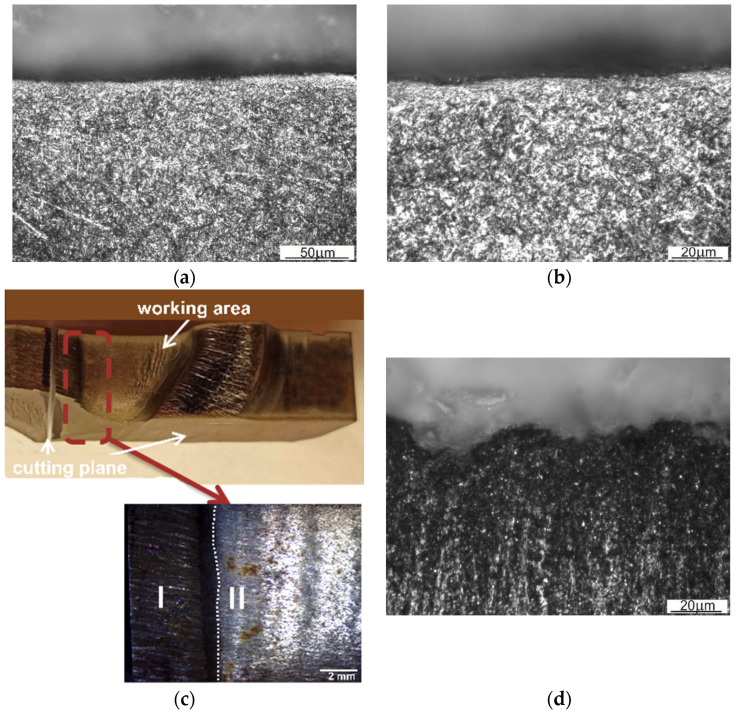
Microstructure of: (**a**) the top layer lying on the surface, having no contact with the wrought forging (area 1a), (**b**) the magnification of area 1a, (**c**) view of a fragment of the insert with inner web (area 2a), and (**d**) microstructure of the top layer in the area of an inner web (area 3a).

**Figure 5 materials-16-00351-f005:**
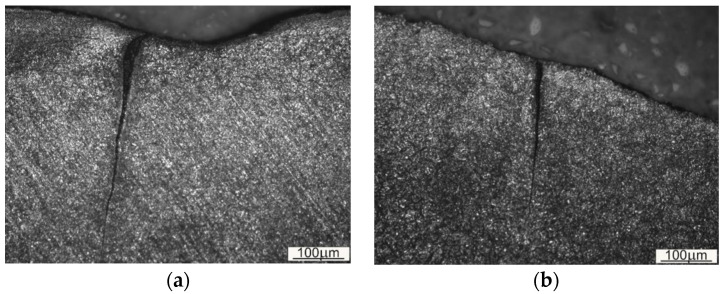
Microstructure of the top layer in the area of the die profile transition into an outer flash web: (**a**) in area 3b1 (Figure 3—area 3b1) and (**b**) in area 3b2 (Figure 3).

**Figure 6 materials-16-00351-f006:**
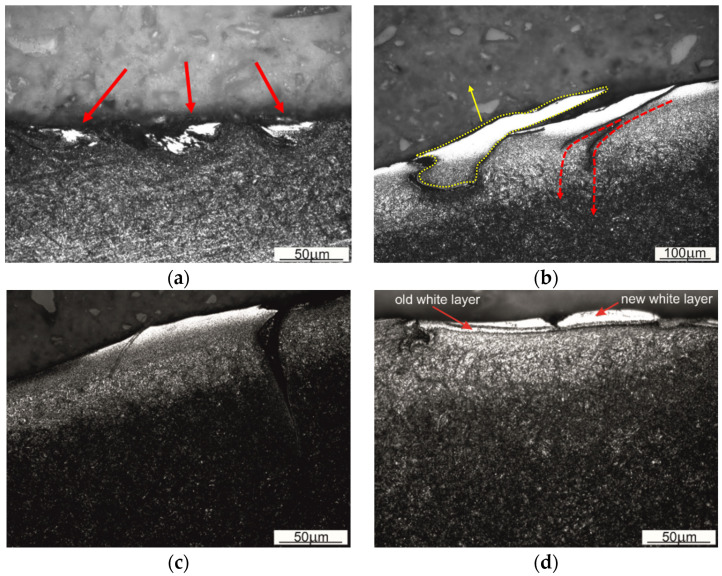
The top layer showing microstructural changes on the surface of the sample in the flash area to the web edge: (**a**) perpendicular (red arrows indicate the white layer), (**b**,**c**) parallel (yellow dotted line indicates the corrugation of the white layer, and red dotted lines indicate the course of further development of the crack formed between the white layer and the rest of the material, and (**d**) parallel at the fillet radius.

**Figure 7 materials-16-00351-f007:**
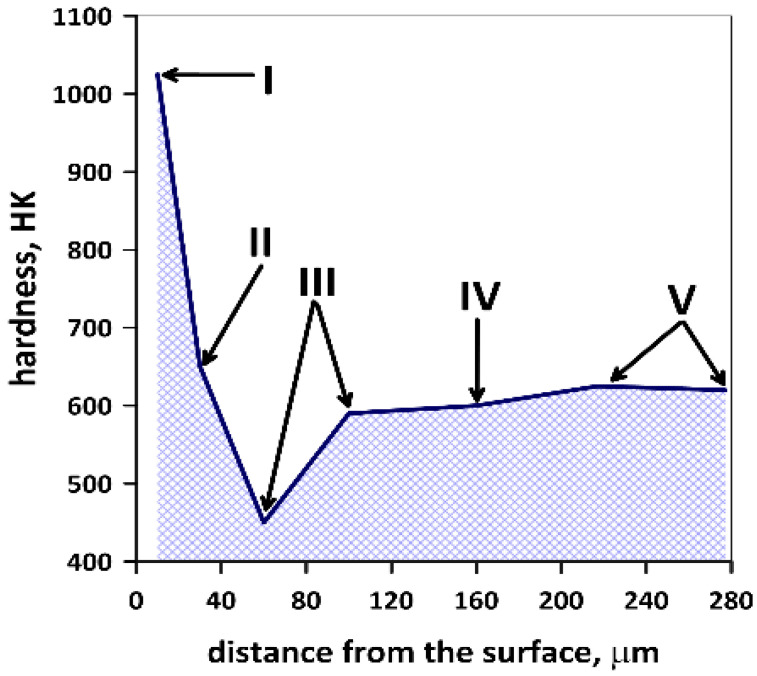
Changes in hardness as a function of the distance from the die insert surface.

**Figure 8 materials-16-00351-f008:**
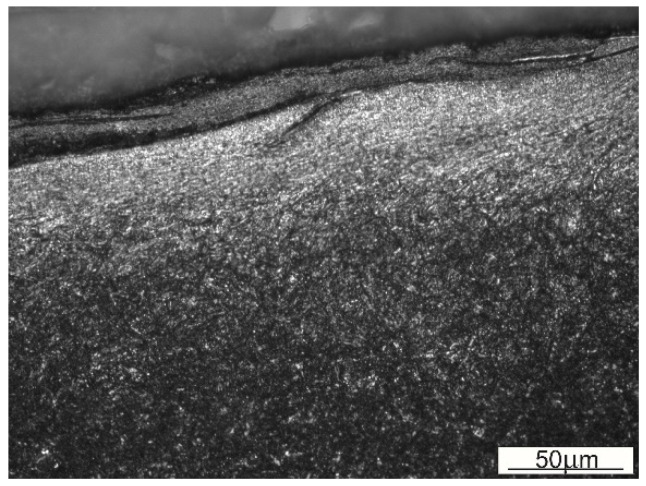
Forged material sticking to the die surface in the area of flash gap.

**Figure 9 materials-16-00351-f009:**
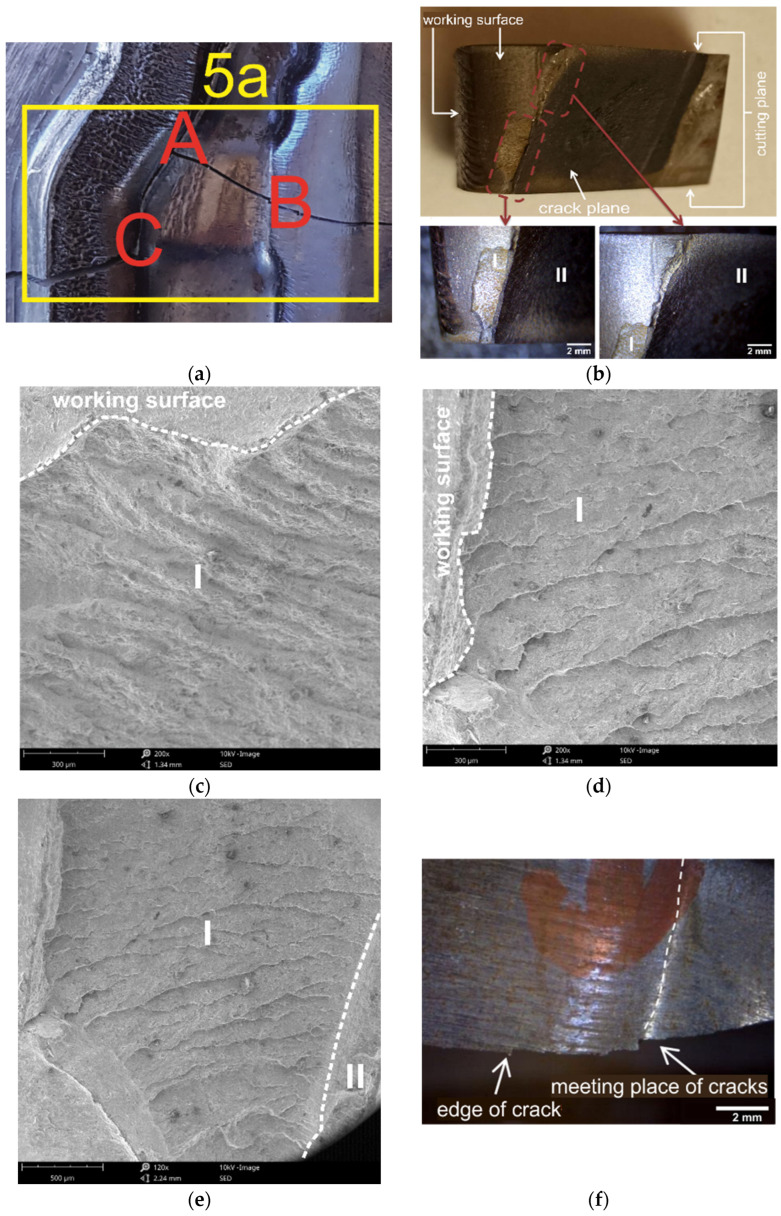
The examined area of the crack in the central part of the die: (**a**) pictorial view, (**b**) the examined area A, (**c**–**e**) pictures of the delamination area, (**f**) the examined area B, (**g**,**h**) notch and change in the plane of breaking crack, and (**i**,**j**) chipping of material in the notch area.

**Figure 10 materials-16-00351-f010:**
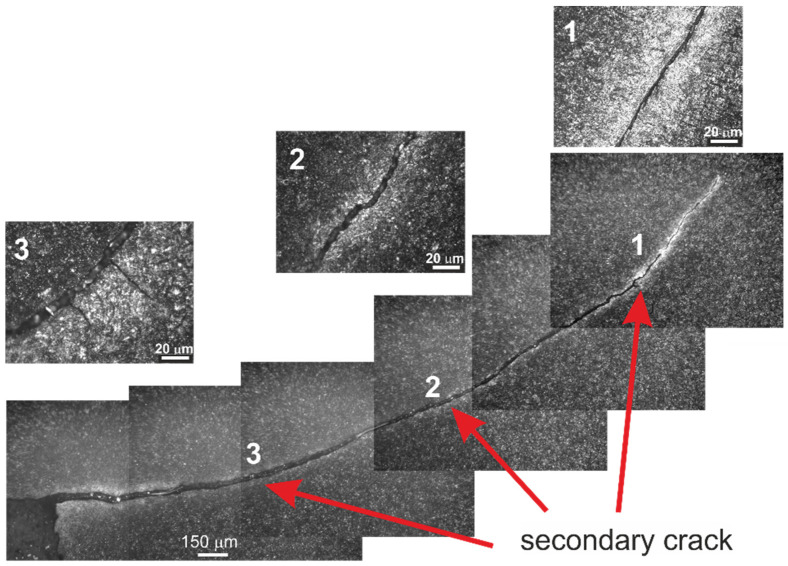
The view of the secondary crack in area C from Figure 5a. 1, 2, 3 indicate the areas that were have been enlarged for better legibility.

**Figure 11 materials-16-00351-f011:**
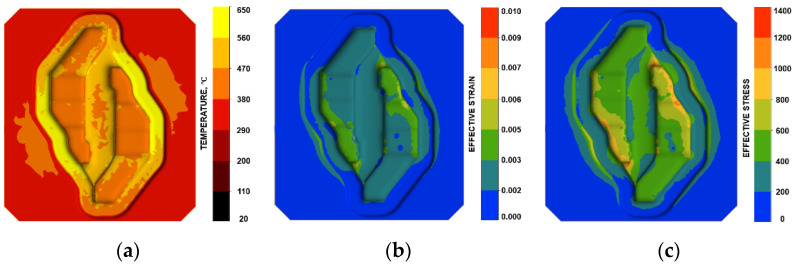
Tool working surface: distribution of (**a**) temperature, (**b**) effective strain, and (**c**) effective stress.

**Figure 12 materials-16-00351-f012:**
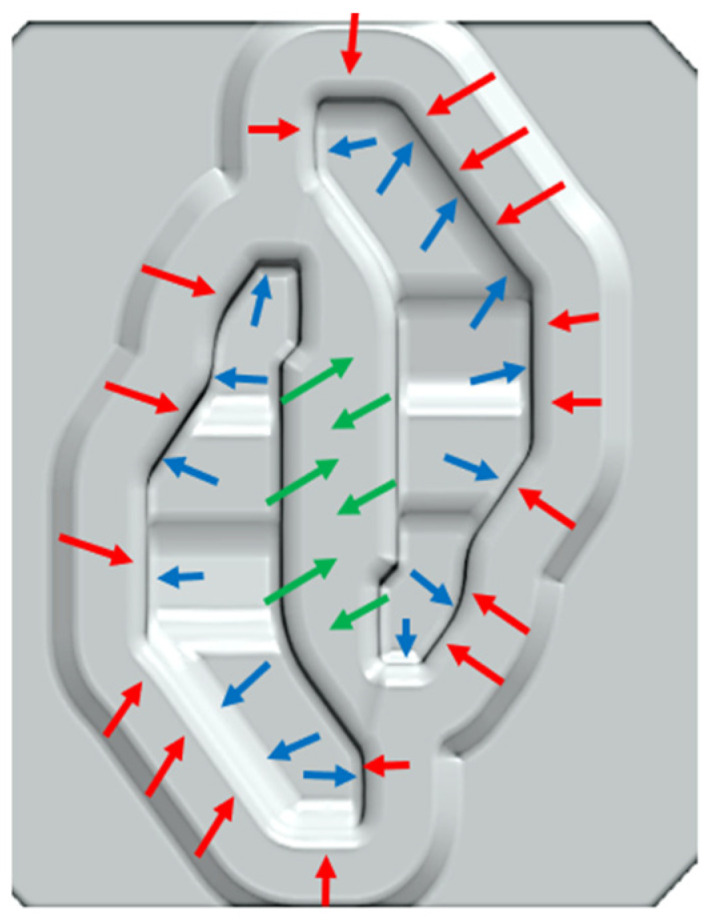
The areas of the dominant wear mechanisms on the surface of the forging tool: erosive tribological (green arrow), fatigue cracking (blue arrow), and white layer formation (red arrow).

**Table 1 materials-16-00351-t001:** Chemical composition (% by mass) of the tool material in relation to the EN ISO 4957: 2018 standard.

	C	Si	Mn	P	S	Cr	Mo	V
Min	0.33	0.8	0.25	–	–	4.8	1.1	0.3
Analysis	0.52	0.22	0.46	0.01	0.011	4.69	2.29	0.51
Max	0.41	1.2	0.50	0.03	0.02	5.5	1.5	0.5

## Data Availability

Data sharing not applicable.

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
