# Peer review of "Wear Mechanisms of the Forging Tool Used in Pre-Forming in a Double Forging System of Truck Parts"

_materials, 2022, doi:10.3390/ma16010351_

Round 1
Reviewer 1 Report
The die life plays an important role in improving the product quality and reducing the processing cost. This study presents an example of the coexistence of multiple wear mechanisms of die in hot forging process. Detailed microstructure examinations were carried out at different areas of formed die. The nucleation cause of fatigue crack and some irregularities in the preparation of die for production process were revealed. The research results can provide theoretical value for practical production. Some comments were listed as follows for a further improvement.
The material of formed parts, and the forming velocity and initial temperature for pre-forming need be provided, which determines the use conditions of the die.
The type and content of element type of surface and white layers at different zones of pre-forming die were suggested to further analyzed, so as to better reveal the wear and fracture mechanisms.
Whether it is possible to provide the quantitative distributions of temperature, stress and elastic-plastic strain of the die during forming process through experimental measurement or numerical simulation, which are closely related to wear and fracture of die.
Author Response
Dear Reviewer,
Thank you for your valuable comments and suggestions, which we included in the text of the manuscript and responded to in a separate file.
Best regards,

Reviewer 2 Report
1. Quality of microstructure is very poor. In most of the microstructures, scale bar is not very clear.
2. Quality of Fig. 7 is very poor and not legible. Replot this figure please.
3. “Fig. 10. The view of secondary crack in area C – 5a” You mean Figure 5a?
Author Response

(The authors gave the same response as above.)

Reviewer 3 Report
The paper needs a major revision and I invite the authors to modify the manuscript according to the following comments.
1. The mechanical properties like tensile test results, hardness, and fracture toughness of the tool and die materials must be included in the manuscript
2. What kind of lubricant was used during the forging process, please include details in the paper.
3. The etchant details and etching procedures must be added to the experimental part
4. Please include EDS and XRD results of the white layer, dark layer, austenite layer, plasticized layer, and the parent material for comparison
Author Response

(The authors gave the same response as above.)

Round 2
Reviewer 3 Report
The current revised version can be accepted for publication